# Research on technological innovation decision-making considering government subsidies and corporate reputation

**Yu Kang, Zhe Huang**⬚*

School of Business Administration, Shenyang Pharmaceutical University, Shenyang, Liaoning, China

* huangzhe2000@sina.com

**Data Availability Statement:** All data files are available from OSF database in https://osf.io/vsz2y/?view_only=fe34edc619dc4b03ad4d102a10a9bf90.

## Abstract

Aiming at the information asymmetry between pharmaceutical enterprises' technological innovation decisions and government subsidy strategy, this paper establishes a differential game model consisting of the government and a single pharmaceutical company, proposes three different government subsidy strategies, and obtains an equilibrium solution with the help of the Hamilton-Jacobi-Bellman equation, taking into consideration of the transmission effect of the enterprise's reputation. First, the innovation decisions of pharmaceutical firms without government subsidies are analysed, and based on this, the optimal strategies with government subsidies for non-cooperative pacts and cooperation between the government and enterprises are analysed separately. In addition, the effects of different subsidy strategies on the government's investment efficiency, corporate reputation, and the choice of corporate innovation strategies are compared, and the results are verified by numerical analysis. Finally, based on the results of the study, references and suggestions are provided for the formulation of government subsidy policies as well as corporate innovation decisions. The results show that: government subsidies can effectively stimulate the innovation ability of pharmaceutical enterprises and improve their reputation; the more sensitive an enterprise's reputation is to the coefficient of technological innovation, the more it can improve the enterprise's innovation level; and the coordination contract of government-enterprise cooperation can realize the Pareto improvement of the benefits of the government and enterprises.

## 1 Introduction

The importance of innovation for the country's high-quality development is self-evident. At the same time, innovation is particularly important for enterprises. It is one of the important strategic means for enterprises to maintain their competitive advantage [1]. Through innovation, enterprises can promote production and business performance, expand the market scale, seize market share, and win a market competitive position. The key to innovation lies in the ability of technological innovation. How to stimulate the technological innovation ability of enterprises has always been a topic of extensive research and discussion by scholars. On the

**Funding:** This work is supported by the research results of Liaoning Economic and Social Development Research Project in 2024 under Grant 2024lslybkt-057. The funders had no role in study design, data collection and analysis, decision to publish, or preparation of the manuscript. There are no ethical or legal restrictions on sharing my data publicly.

**Competing interests:** The authors have declared that no competing interests exist.

one hand, in the process of innovation and development in China, there are problems such as financing difficulties, lack of funds, and lack of innovative talents in enterprise innovation [2]. Given the lack of technological innovation resources, government subsidies have become a supplement to free external resources. Based on the resource-based theory, government subsidy support can fill the gap of insufficient innovation resources of enterprises, compensate for the cost of carrying out innovation activities, weaken the financing constraints of enterprises, and improve the enthusiasm for innovation. As an important policy tool to stimulate the improvement of enterprise innovation level, government subsidies have always been a hot issue in the field of innovation research [3]. Different subsidy strategies adopted by the government have different effects on the level of technological innovation of enterprises. Wang, Nan [4] measured the technological innovation value of the project under different subsidy methods (subsidy time, subsidy form, subsidy amount, etc.), and discussed the different effects of different subsidy methods on the project. Determine the optimal subsidy method for technological innovation activities, so that limited government subsidies can be effectively utilized. Therefore, this paper considers the impact of government subsidies on corporate technological innovation decision-making has certain practical significance.

On the other hand, in the process of technological innovation, the problem of information asymmetry has become increasingly prominent. Aiming at the ' adverse selection ' and ' moral hazard ' caused by the information asymmetry between investors and operators, managers and employees, signal transmission can timely balance the information acquisition between the two parties and minimize the damage to the interests of both parties. Based on the signal transmission theory, technological innovation of enterprises is a manifestation of fulfilling social responsibility [5]. As a transmission signal, Corporate Social Responsibility (CSR) can convey useful information to corporate stakeholders to actively carry out technological innovation, reduce the information asymmetry between enterprises and external stakeholders, and thus reduce the investment risk of external stakeholders [6–11]. Numerous studies have shown that corporate social responsibility can improve the performance of enterprises, enhance the reputation of enterprises, and promote the sustainable development of enterprises [12, 13]. Based on this, it is assumed that the fulfillment of social responsibility based on technological innovation is also conducive to enhancing the reputation value of enterprises. The behavior of technological innovation of enterprises can be fed back by corporate reputation. Good corporate reputation will attract more social support and capital investment. As the most important external stakeholders of enterprises, the government can judge the technological innovation behavior of enterprises according to the reputation of enterprises so as to reduce information asymmetry [14]. This is also an important reason for this paper to introduce reputation as a dynamic variable of research.

Pharmaceutical enterprises are an important pillar of national life safety, shouldering the mission and responsibility of safeguarding people's health. However, from the current status of innovation development in the pharmaceutical industry, due to the high investment in innovative drug development, high technical barriers, long transformation cycle and high risk, therefore, some pharmaceutical companies are less motivated to choose high-risk and high-yield innovative drug research and development [15]. In order to respond to the strategic goal of ' Healthy China 2030 ', improve the health literacy of the whole people, and eliminate the harm of a large number of major diseases while ensuring the safety of drugs, this requires the joint efforts of the whole society. As an important member, pharmaceutical companies should assume corresponding social responsibilities. Therefore, this paper selects pharmaceutical enterprises as the key research object, focusing on the impact of different government subsidy strategies on the technological innovation decision-making of pharmaceutical enterprises under information asymmetry, in order to solve the following problems:

1. How different government subsidy strategies affect the level of firms' innovation effort, the level of firm reputation and the profits of pharmaceutical firms.

2. How do changes in corporate reputation affect the innovation decisions of pharmaceutical firms?

3. What is the optimal government subsidy strategy that enhances the efficiency of government subsidies?

The main contributions of this paper: Firstly, it explores the efficiency of government subsidies under information asymmetry and its impact on the technological innovation level of enterprises, verifies the existence of a promotion effect between government subsidies and enterprise innovation, and provides a reference for the government to formulate innovation subsidy policies; secondly, it analyses the trend of changes in the level of corporate reputation under different circumstances, and studies how the level of corporate reputation influences the technological innovation decision-making of enterprises, and provides a reference for the enterprises to formulate the appropriate level of technological innovation efforts. Finally, the feedback evaluation function of consumers is verified, and the degree of consumer demand for new drugs has a certain influence on the proportion of government subsidies, and the greater the demand, the greater the proportion of subsidies that the government is willing to bear. The findings of the study are intended to provide reference suggestions for pharmaceutical companies to develop and gain higher profits in the long run and for the government to improve the efficiency of subsidies, which is of strong practical significance for stimulating the innovation vitality of the pharmaceutical industry, accelerating the high-quality development of the pharmaceutical industry, and maximizing the welfare of the society.

The rest of this paper is organized as follows: Section 2 is a literature review. By reading a large number of literature, it summarizes the importance and practical significance of studying government subsidies and corporate reputation on corporate technological innovation decision-making, indicating that the inclusion of these two research variables has a certain research basis. Section 3 is the problem description and basic assumptions, including the definition of related variable symbols; the fourth section is to build a differential game model; section 5 is a comparative analysis of the equilibrium solutions of the three models; section 6 is numerical simulation to observe the changing trend of relevant indicators and test the model. Section 7 is the conclusion and policy implications, summarizes the research contributions and deficiencies, and proposes future research directions.

## 2 Literature review

The research in this paper is closely related to the related literature in two aspects: first, the study of firms' technological innovation decisions under government subsidy policies; and second, the study of firms' technological innovation and corporate reputation.

In terms of research related to enterprise technology innovation decision-making under government subsidy policy. There have been numerous studies on the impact of government subsidies on enterprise innovation, and the trend is on the rise [16]. The conclusions of the current academic research on the relationship between the two have not been unified. Most of these studies adopt an empirical research paradigm and use panel data to examine the impact of government subsidies on firms' innovation efforts. Most of the studies concluded that government subsidies have a positive effect on firms' innovation and help firms gain a competitive advantage [17]. Junhua Wang [1], Lili Jia [18], Xucheng Wu [19], and Shuwei Chen [20] conducted empirical studies based on data from listed companies and concluded that government subsidies can effectively promote enterprise R&D investment and output. However, a part of

the scholars believe that government subsidies have no significant impact on technological innovation [21, 22], and even have a crowding out effect [23], which is not conducive to the development of enterprise innovation. Some other scholars believe that there is a threshold effect between the two and that when government subsidies exceed a certain threshold, they will inhibit firms' innovative performance [24–27]. The inconsistency observed in these studies may stem from the limitations of panel data analysis in elucidating in depth the decision-making logic at the micro level of each actor [28]. Currently, there are also works on related research using behavioral decision-making models. These studies tend to emphasize exploring the optimal decision-making of various government subsidy policies for different players in the supply chain [14, 29]. Yipeng Lan [30] et al. explored the impact of government subsidy strategies on decision-making variables of new drug discovery and development (R&D), the efficiency of pharmaceutical firms as well as the welfare of the society, taking into account the intra-industry horizontal spillovers and the inter-industry vertical spillovers. Li Xufeng [31] et al. investigated the interactions among manufacturers' smart product innovation decisions, platform transfer payments, and government subsidies by constructing a game-theoretic analytical model. Various studies usually introduce different factors (e.g., consumer preference [32], product eco-friendliness [33], etc.) into the analysis. Established studies focus on the impact of government subsidies on technological innovation decisions, but less on the efficiency of government subsidies. This study constructs a differential game model from a dynamic perspective and analyses the optimal choice of innovation subsidy strategies in the context of information asymmetry from both government and firm perspectives, which has rarely been considered in previous work.

Research aspects of corporate technological innovation and corporate reputation. Innovative ability is an important source of corporate reputation, and the corporate reputation evaluation system of many international and domestic authoritative business magazines, such as the U.S. Fortune magazine, includes innovation. Existing literature has focused on the reputational effects of innovation capability. DAVID HAL HENARD [34] demonstrated through Meta-analysis that market performance determinants such as product innovativeness and technological synergies enable firms to achieve a high reputation for product innovation, and thus a higher level of customer loyalty than competitors who lack innovation. Juan-Gabriel Cegarra- Navarro [35] found through a survey study that technology assimilation supported by green skills not only avoids or mitigates the effects of embarrassment-induced stress, anxiety and fear, but also supports organizational reputation. Zhang Qingfei [36] argued that an enterprise's technological innovation capability has a signaling effect, and stakeholders who pounce on the signal of an enterprise's innovation capability will form an overall evaluation of the enterprise, which will lead to the formation of the enterprise's reputation, and confirmed through empirical research that the enterprise's reputation is supported by the enterprise's external resources from the point of view of the enterprise's external resources acquisition (customers' purchasing rate and loyalty, investors' favor and investment, the government's policy inclination and financial support, the supplier's lenient purchasing policies and prices, and the quality and loyalty of internal employees) will have a positive impact on corporate performance. Yonni Angel Cuero-Acosta's [37] study explains how MSMEs build their reputation to influence firm performance through innovation and knowledge accumulation. However, established studies have not explored the mediating mechanism between innovation and firm performance from a reputation perspective. In addition, there are insufficient and controversial studies on the impact of reputation on firms' technological innovation. Muhammad Waqas et al. [38] used structural equation modeling to measure and validate that corporate reputation has a moderating effect on supply chain innovation. Wenxiu Hu et al. [39] however, do not support the reputation hypothesis, arguing that the information effect plays a dominant

role; that CSR disclosure can mitigate the relationship between managers and investors, controlling shareholders and minority shareholders, and alleviate the problem of financing constraints, thus improving the sustainability of innovation. Xiaoya Han [40] explores product pricing strategy and quality strategy by considering the brand reputation of the innovating firms and the quality expectations of consumers. The results show that the optimal price and quality level of an innovative product are not always positively related to the brand goodwill of the innovative firm. In addition, facing consumers who are insensitive to quality bias, innovative firms with low brand goodwill can hardly gain an advantage in quality strategy. Wu Xiaojie et al. [41] constructed a differential game model in which a duo-oligopoly firm competes for market share through goodwill accumulation and product quality improvement activities; and investigated the optimal advertising and product quality competition strategies of duo-oligopoly firms through goodwill to attract new customers and product quality as a way to establish and maintain the loyalty of target customers. Based on existing research, this paper incorporates the reputation variable into the differential game model as a dynamic variable to explore the mechanism of reputation's influence on corporate technological innovation and its mediating mechanism between innovation and corporate performance.

Existing research on enterprise technological innovation has provided us with a compendium of factors influencing enterprise technological innovation, and, at the same time, pointed out the problems of high innovation costs and inefficient government subsidies for pharmaceutical companies. All these help us to gain relevant basic knowledge and understand the real context. Research on government subsidy strategies and studies related to corporate reputation provide the necessary technical support for the quantification of subsidy strategies and the proposal of subsidy methods, the establishment of game models and the selection of calculation methods [33, 42–46]. However, the existing studies do not propose subsidy strategies with guiding significance for promoting the development of technological innovation in pharmaceutical companies, and there is a relative lack of research on corporate reputation on corporate innovation performance. It should be noted that reputation, as an important intangible asset of enterprises, is particularly important for enterprises, especially in the pharmaceutical industry. Therefore, different from previous studies, this paper constructs three different subsidy strategies from the perspective of corporate reputation, and through comparative analysis, explores the impact of different government subsidies on the technological innovation level of pharmaceutical companies to provide a reference for the behavioral decision-making of pharmaceutical companies and the government. The innovativeness of this paper is mainly reflected in the following aspects: first, a series of conduction relationships are established through the dissemination of reputation. An enterprise's technological innovation efforts affect the enterprise's reputation → the enterprise's reputation affects consumer demand → the demand further affects the enterprise's profit. By modeling this conduction relationship, a mathematical description of the problem is formed. Second, from the government's standpoint, the advantages and disadvantages of various participation methods are examined for the purpose of reducing information asymmetry and obtaining optimal subsidy efficiency. Thirdly, from the perspective of the enterprise, appropriate technological innovation decisions are made based on the subsidy strategy developed by the government in order to maintain a good corporate reputation and further attract government support to incentivize corporate innovation, forming a virtuous circle and achieving sustainable development.

## 3 Preliminaries

In this section, the symbol definition、 problem description and research hypothesis are introduced below.

### 3.1 Symbol definition

In order to increase the readability of the article, the key symbols and definitions are listed in Table 1.

### 3.2 Problem description and model building

Consider a simple strategic system consisting of a government and a single pharmaceutical company in continuous time t∈ (0, ∞). In this system, pharmaceutical companies take social responsibility by producing higher-quality drugs through technological innovation. Technological innovation will bring high-quality products to the consumer market, thus attracting the attention of consumers and further improving the reputation of enterprises, and a good reputation will bring customer loyalty and brand value [9], thus helping enterprises to expand market share and occupy the market quickly, obtain certain competitive advantages, and bring certain profits to enterprises. Since technological innovation consumes a lot of costs, in order to make up for part of the cost losses of pharmaceutical enterprises, the government will provide corresponding R&D subsidies. In order to reduce the agency problems such as adverse selection and moral hazard caused by information asymmetry, the government will make corresponding subsidy decisions through the positive signal of enterprises' social responsibility. This paper constructs a simple strategy system consisting of two participants, the government and a single pharmaceutical company implementing technological innovation CSR. Consider the game problem between them in continuous and dynamic time. In this paper, the government is chosen as the leader of the differential game model, and the pharmaceutical enterprise is chosen as the follower, which constitutes a simple Stackelberg differential game model. Combined with the research background, this paper constructs a game decision-making mechanism for government, pharmaceutical enterprises and consumers to determine the level of technological innovation, as shown in Fig 1. The general order of the game is: First of all, the government determines the subsidy coefficient according to the previous reputation level of pharmaceutical enterprises and consumers' attitudes towards the technological innovation level of pharmaceutical enterprises. Secondly, pharmaceutical enterprises determine the input cost coefficient of technological innovation on the basis of the subsidy coefficient given by the government. At the same time, consumers have the evaluation and feedback function on the

**Table 1. Symbols and definitions of main variables in this paper.**

| Symbol | Definition |
|---|---|
| $\alpha$ | Cost coefficient of technological innovation effort in pharmaceutical enterprises |
| $K(t)$ | Technological innovation level of pharmaceutical enterprises |
| $\lambda$ | The influence coefficient of technological innovation on corporate reputation of pharmaceutical enterprises |
| $\pi$ | The influence coefficient of technological innovation of pharmaceutical enterprises on political benefit of government |
| $\pi_G$ | Marginal revenue for the government |
| $\pi_M$ | Marginal profits for pharmaceutical companies |
| $\rho$ | Discount rates for governments and pharmaceutical companies |
| $Q(t)$ | Demand function for technological innovation in pharmaceutical enterprises |
| $T(t)$ | The reputation of pharmaceutical companies |
| $\delta$ | The rate of reputational decay, also known as the rate of decay |
| $\theta$ | The influence coefficient of technological innovation on demand of pharmaceutical enterprises |
| $\beta$ | The influence coefficient of corporate reputation on demand |
| $F(t)$ | Coefficient of government subsidy |

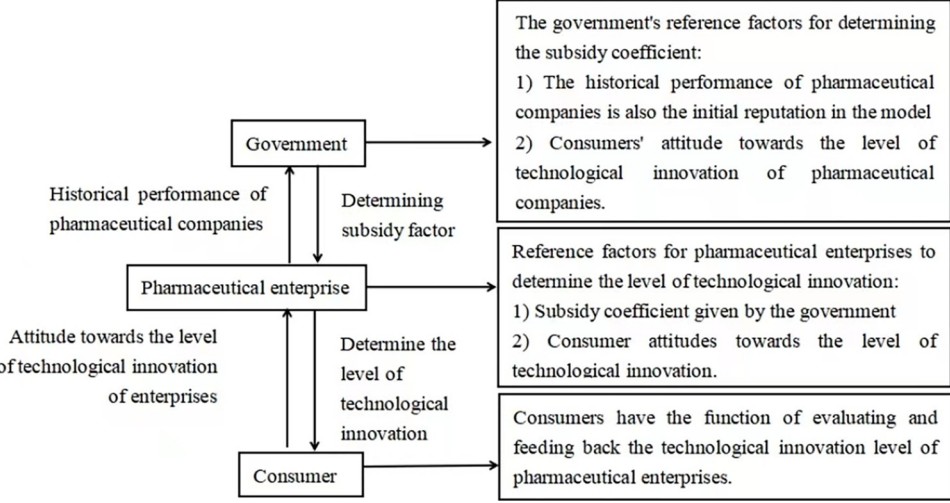

**Fig 1. Game decision-making mechanism.**

technological innovation level of pharmaceutical enterprises. Pharmaceutical companies should also pay attention to the changes in consumers' attitudes in real-time, so that their innovation level and reputation changes can be maintained in a more ideal state.

### 3.3 Fundamental assumption

**Assumption 1.**   Existing studies [47, 48] all hold that the cost of technological innovation is a quadratic function of its level of technological innovation, assuming that the cost of technological innovation of a pharmaceutical enterprise is related to its level of technological innovation, at time t, the cost of technological innovation of a pharmaceutical enterprise is, where k (t) is the level of technological innovation of an enterprise, and k (0) represents the initial level of innovation is 0. $\alpha$ is the cost coefficient of enterprise technological innovation input, and $\alpha > 0$.

**Assumption 2.**   The reputation level of enterprises is positively correlated with the technological innovation level of enterprises. With the increasing market demand for specific drugs, pharmaceutical enterprises pay more and more attention to the innovative research and development of specific drugs. The better consumers' recognition of pharmaceutical enterprises fulfilling the social responsibility of technological innovation, the reputation of enterprises will gradually improve. With the increasing intensity of market competition, the speed of technological update and iteration will gradually accelerate and replace the original technical level, and the reputation level of enterprises will also decline and may be submerged. According to the random reputation model proposed by Chen Jingquan [49] and Zhao Liming [50], the decline of the reputation of enterprises over time. Its dynamic change process can be expressed as a random differential equation.

$$\bullet T(t) = T_0 + \lambda k(t) - \delta T(t) \tag{1}$$

Where T (t) represents the reputation of the company, $\lambda > 0$ represents the influence coefficient of the level of technological innovation on the reputation of the company, $\delta > 0$ represents the decline rate of corporate reputation over time due to the update iteration of technology when the company does not make innovation efforts, and $T_0 = T(0)$ represents the initial reputation of the pharmaceutical company.

**Assumption 3.** Assuming that consumers' purchasing behavior is jointly affected by the level of corporate reputation and technological innovation, enterprises expand consumer demand through "cause-related marketing" and improving drug quality. According to Zhao Liming's research hypothesis, it is assumed that the demand function of pharmaceutical enterprises is

$$Q(t) = \beta T(t) + \theta k(t) \tag{2}$$

Where $\beta$ is the influence coefficient of enterprise reputation on consumer demand, $\theta$ is the influence coefficient of enterprise technological innovation level on demand, and $\beta > 0$, $\theta > 0$.

**Assumption 4.** Firms' innovation has a reputational effect, and governments usually select reputable firms for innovation subsidies. Governments incentivize innovation in a variety of ways, such as cash incentives and tax breaks. The government's subsidy coefficient is related to the cost of pharmaceutical companies' technological innovation efforts [51]. Referring to the assumptions in the literature [49], this paper regards government subsidies as cash subsidies, and F (t) is the government subsidy coefficient, which satisfies $0 \leq F(t) < 1$. When F (t) = 0, it means that the government does not provide subsidies. F (t) < 1 indicates that the government only bears a part of the innovation cost of the enterprise.

**Assumption 5.** Suppose that the government and the pharmaceutical industry have the same positive discount rate $\rho$, and the goal of both is to maximize their interests in the infinite positive range. The benefits of the government should include two parts: one part comes from the technological innovation of pharmaceutical enterprises to bring more welfare to patients, which belongs to the direct social benefits of the government, and the other part comes from the technological innovation of pharmaceutical enterprises to expand consumer demand and help enterprises and the government to obtain corresponding indirect economic benefits. Therefore, the objective functions of the two are respectively as follows:

$$\max_F J_G = \int_0^\infty e^{-\rho t}[\pi_1 k(t) + \pi_G Q(t) - F(t)C(t)]\mathrm{d}t \tag{3}$$

$$\max J_M = \int_0^\infty e^{-\rho t}[\pi_M Q(t) - (1 - F(t))C(t)]\mathrm{d}t \tag{4}$$

Where $\pi_1$ is the influence coefficient of technological innovation of pharmaceutical enterprises on government social benefits, and also indicates the degree of patients' demand for new drugs; $\pi_G$ and $\pi_M$ represent the marginal income of the government and pharmaceutical enterprises respectively, reflecting the economic benefits of the two, and $\pi_1$, $\pi_G$, $\pi_M > 0$.

## 4 Model analysis

In this paper, there are only two control variables k (t) and F (t), and one state variable T (t). For the convenience of calculation, the time unit t is omitted in the writing of reference [52] to obtain the feedback equilibrium under the static strategy. Based on the previous description and hypothesis, this part further analyzes the game model of three different situations.

### 4.1 Non-cooperative contract model of anarchic subsidies(N)

In order to analyse the effects of the government subsidy policy, the results of the no-cost-sharing scenario are first given as a benchmark for comparing the effects of the government subsidy policy. Since the government does not give any subsidy to the enterprise at this time, the subsidy coefficient F (t) = 0, and the enterprise bears the cost of technological innovation alone, and the superscript N represents the situation of no government subsidy, the objective

function of the government and the enterprise at this time is as follows:

$$\max J_G^N = \int_0^\infty e^{-\rho t}[\pi_1 k + \pi_G Q]\mathrm{d}t \tag{5}$$

$$\max J_M^N = \int_0^\infty e^{-\rho t}[\pi_M Q - C]\mathrm{d}t \tag{6}$$

**Theorem 1** In the static game of non-cooperative contracts with no government subsidies, the Stackelberg equilibrium strategy of pharmaceutical enterprises is as follows:

$$T(t)^N = (T_0 + \frac{Z^N}{Y^N})e^{Y^N t} - \frac{Z^N}{Y^N} \tag{7}$$

Where $Y^N = -\delta$、 $Z^N = \frac{\lambda \pi_M \theta(\rho+\delta)+\pi_M \beta \lambda^2}{\alpha(\rho+\delta)}$

$$k^N = \frac{\pi_M \theta(\rho + \delta) + \pi_M \beta \lambda}{\alpha(\rho + \delta)} \tag{8}$$

It is proved that the optimal benefit function Vi (T), i∈ (G, M) of the government and pharmaceutical enterprises exists HJB equation for any T≥0 according to the optimal control theory:

$$\rho V_G^N = \max\{\pi_1 k + \pi_G Q + V_G^{N'}(\lambda k - \delta T)\} \tag{9}$$

$$\rho V_M^N = \max\{\pi_M Q - C + V_M^{N'}(\lambda k - \delta T)\} \tag{10}$$

Eq (10) is obtained by taking the partial derivative of the first order concerning k and setting it to 0

$$k = \frac{\pi_M \theta + V_M^{N'}\lambda}{\alpha} \tag{11}$$

By substituting Formula (11) into Formula (9) and Formula (10), we can get:

$$\rho V_G^N = (\pi_G \beta - \delta V_G^{N'})T + (\pi_1 + \pi_G \theta + V_G^{N'}\lambda)\frac{\pi_M \theta + V_M^{N'}\lambda}{\alpha} \tag{12}$$

$$\rho V_G^N = (\pi_M \beta - V_M^{N'}\delta)T + (V_M^{N'}\lambda + \pi_M \theta)\frac{\pi_M \theta + V_M^{N'}\lambda}{\alpha} - \frac{(\pi_M \theta + V_M^{N'}\lambda)^2}{2\alpha} \tag{13}$$

It can be seen from Eqs (12) and (13) that the optimal benefit function of T is the solution of the HJB equation, and the specific expression of function $V_i$ (t),i∈ (G, M) is as follows:

$$V_G = u_1 T + u_2、 \quad V_M = r_1 T + r_2 \tag{14}$$

Where $u_1$, $u_2$, $r_1$ and $r_2$ are all constants, substituting Eq (14) and its derivative with respect to T into Eq (12) (13) yields:

$$\rho(u_1 T + u_2) = (\pi_G \beta - \delta u_1)T + (\pi_1 + \pi_G \theta + u_1 \lambda)\frac{\pi_M \theta + r_1 \lambda}{\alpha} \tag{15}$$

$$\rho(r_1 T + r_2) = (\pi_M \beta - r_1 \delta)T + (r_1 \lambda + \pi_M \theta)\frac{\pi_M \theta + r_1 \lambda}{\alpha} - \frac{(\pi_M \theta + r_1 \lambda)^2}{2\alpha} \tag{16}$$

Eqs (15) and (16) are satisfied for all T≥0, and the parameters of the optimal benefit function can be obtained by comparing the coefficients of the same terms at the left and right ends:

$$u_1 = \frac{\pi_G \beta}{\rho + \delta}$$

$$u_2 = \frac{[\pi_M \theta(\rho + \delta) + \pi_M \beta \lambda][(\pi_1 + \pi_G \theta)(\rho + \delta) + \pi_G \beta \lambda]}{\alpha \rho (\rho + \delta)^2}$$

$$r_1 = \frac{\pi_M \beta}{\rho + \delta}$$

$$r_2 = \frac{[\pi_M \beta \lambda + \pi_M \theta(\rho + \delta)]^2}{2\alpha \rho (\rho + \delta)^2}$$

By substituting the obtained parameters back to Eq (14), the maximum benefit functions of the government and pharmaceutical enterprises can be obtained as follows:

$$V_G^{N^*} = \frac{\pi_G \beta}{\rho + \delta} T + \frac{[\pi_M \theta(\rho + \delta) + \pi_M \beta \lambda][(\pi_1 + \pi_G \theta)(\rho + \delta) + \pi_G \beta \lambda]}{\alpha \rho (\rho + \delta)^2} \tag{17}$$

$$V_M^{N^*} = \frac{\pi_M \beta}{\rho + \delta} T + \frac{[\pi_M \beta \lambda + \pi_M \theta(\rho + \delta)]^2}{2\alpha \rho (\rho + \delta)^2} \tag{18}$$

By substituting Eq (14) back to Eq (11), the equilibrium solution of pharmaceutical enterprise can be obtained:

$$k^N = \frac{\pi_M \theta(\rho + \delta) + \pi_M \beta \lambda}{\alpha(\rho + \delta)}$$

By substituting the equilibrium solution into the equation of state, the optimal trajectory of reputation can be obtained as:

$$T(t)^N = (T_0 + \frac{Z^N}{Y^N}) e^{Y^N t} - \frac{Z^N}{Y^N}$$

Where $Y^N = -\delta$、 $Z^N = \frac{\lambda \pi_M \theta(\rho + \delta) + \pi_M \beta \lambda^2}{\alpha(\rho + \delta)}$ Theorem 1 has been proved.

**Inference 1**: In the case of non-government subsidies, the technological innovation level of pharmaceutical enterprises is negatively correlated with the discount rate ρ, the reputational decline rate δ and the technological innovation cost coefficient α. It is positively correlated with the coefficient λ of technological innovation level on reputation, the coefficient β of reputation on consumer demand, the coefficient θ of technological innovation level on demand and the marginal income $\pi_M$ of pharmaceutical enterprises.

The conclusion is basically consistent with the reality. Pharmaceutical enterprises can produce and develop more high-quality drugs through technological innovation. For rare patients, they pay special attention to the special drugs in the market. With the attention and emphasis of the society on the use of drugs for rare patients, the social responsibility signal of pharmaceutical enterprises based on technological innovation becomes particularly important and is reflected in the reputation of enterprises. At the same time, the special drugs produced by

pharmaceutical enterprises through technological innovation can greatly arouse the attention of the masses, so that the drug demand $\beta$ and $\theta$ of the pharmaceutical enterprises will increase. When pharmaceutical companies enjoy a certain level of reputation, they will be more proactive in assuming the responsibility for technological innovation, and encourage enterprises to continue to increase the degree of technological innovation.

In addition, it can be seen from inference 1 that the technological innovation level of pharmaceutical enterprises is negatively affected by the cost of technological innovation. The higher the cost, the lower the technological innovation level of enterprises, so the cost-sharing effect of government subsidies will be particularly important. The technological innovation level of pharmaceutical enterprises is only related to their own marginal income, which has nothing to do with the marginal income of the government, indicating that pharmaceutical enterprises make decisions to maximize their own interests, without considering the interests of the whole system.

## 4.2 Non-cooperative contract model of government subsidies(H)

This model considers the case of government subsidies, that is, F≠0. In order to encourage pharmaceutical enterprises to carry out technological innovation, the government will give certain financial subsidies to share part of the cost of technological innovation. Here, technological innovation is dominated by pharmaceutical enterprises, so the government is the leader and the pharmaceutical companies are the followers. In the long run, the government and pharmaceutical enterprises constitute a Stackelberg differential game. Due to the information asymmetry situation between the government and the enterprise, the government cannot observe the innovative technology level of the enterprise, and as a limited rational government, it will make decisions based on the reputation level of the enterprise [53]. The two play a Nash non-cooperative game, where both parties make decisions simultaneously and independently, the government decides its own government subsidy coefficient, and the pharmaceutical enterprise decides its own level of technological innovation, in order to realise their respective maximum interests. The game decision is divided into two stages: in the first stage, the government determines its subsidy coefficient for cost sharing; and in the second stage, the pharmaceutical companies decide their technological innovation level based on the subsidy given by the government. Both parties make decisions with the goal of maximizing their respective benefits. The superscript H represents the situation with government subsidies, and the objective functions in this case are as follows:

$$\max J_G^H = \int_0^\infty e^{-\rho t}[\pi_1 k + \pi_G Q - FC]\mathrm{d}t \tag{19}$$

$$\max J_M^H = \int_0^\infty e^{-\rho t}[\pi_M Q - (1-F)C]\mathrm{d}t \tag{20}$$

**Theorem 2** When the government gives a certain proportion of financial subsidies, the game equilibrium results between the government and pharmaceutical enterprises are as follows:

$$T(t)^H = \left(T_0 + \frac{Z^H}{Y^H}\right)e^{Y^H t} - \frac{Z^H}{Y^H} \tag{21}$$

Where $Y^H = -\delta$、 $Z^H = \frac{2\lambda[(\pi_1+\pi_G\theta)(\rho+\delta)+\pi_G\beta\lambda]+\lambda[\pi_M\theta(\rho+\delta)+\pi_M\beta\lambda]}{2\alpha(\rho+\delta)}$

$$k^H = \frac{2[(\pi_1+\pi_G\theta)(\rho+\delta)+\pi_G\beta\lambda]+[\pi_M\theta(\rho+\delta)+\pi_M\beta\lambda]}{2\alpha(\rho+\delta)} \tag{22}$$

$$F = \frac{2[(\pi_1 + \pi_G\theta)(\rho + \delta) + \pi_G\beta\lambda] - [\pi_M\theta(\rho + \delta) + \pi_M\beta\lambda]}{2[(\pi_1 + \pi_G\theta)(\rho + \delta) + \pi_G\beta\lambda] + [\pi_M\theta(\rho + \delta) + \pi_M\beta\lambda]} \tag{23}$$

It is proved that the optimal control problem of pharmaceutical enterprise is solved by backward induction method, and the optimal benefit function satisfies the HJB equation:

$$\rho V_M^H = \max\{\pi_M Q - (1 - F)C + V_M^{H'}(\lambda k - \delta T)\} \tag{24}$$

Similarly, the first-order derivation of Formula (24) about the technological innovation level k of pharmaceutical enterprises can be obtained:

$$k = \frac{\pi_M\theta + V_M^{H'}\lambda}{(1 - F)\alpha} \tag{25}$$

In order to maximize its own interests, the optimal function of the government satisfies the HJB equation:

$$\rho V_G^H = \max\{\pi_1 k + \pi_G Q - FC + V_G^{H'}(\lambda k - \delta T)\} \tag{26}$$

Substitute Eq (25) into Eq (26), and find the first derivative of the subsidy coefficient F about the government as follows:

$$F = \frac{2(\pi_1 + \pi_G\theta + V_G^{H'}\lambda) - (\pi_M\theta + V_M^{H'}\lambda)}{2(\pi_1 + \pi_G\theta + V_G^{H'}\lambda) + (\pi_M\theta + V_M^{H'}\lambda)} \ (2A > B); \ F = 0 \ (2A < B) \tag{27}$$

Where $A = \pi_1 + \pi_G\theta + V_G^{H'}\lambda$, $B = \pi_M\theta + V_M^{H'}\lambda$ By substituting Formula (25) and (27) into Formula (24) and (26), it can be simplified:

$$\rho V_G^H = (\pi_G\beta - V_G^{H'}\delta)T + \frac{(2A + B)^2}{8\alpha} \tag{28}$$

$$\rho V_M^H = (\pi_M\beta - V_M^{H'}\delta)T + \frac{B(2A + B)}{4\alpha} \tag{29}$$

According to the structure of Eq (28) and Eq (29), it can be seen that the optimal benefit function of T satisfies the HJB equation. Assumption that:

$$V_G^H = a_1 T + a_2, \ V_M^H = b_1 T + b_2 \tag{30}$$

Where $a_1$、$a_2$、$b_1$、$b_2$ all are constants, and the coefficients of similar terms can be obtained by analogy with both sides of Eqs (28) and (29):

$$a_1 = \frac{\pi_G \beta}{\rho + \delta}$$

$$a_2 = \frac{[2(\pi_1 + \pi_G \theta)(\rho + \delta) + 2\pi_G \beta \lambda) + \pi_M \theta(\rho + \delta) + \pi_M \beta \lambda]^2}{8\rho\alpha(\rho + \delta)^2}$$

$$b_1 = \frac{\pi_M \beta}{\rho + \delta}$$

$$b_2 = \frac{[2(\pi_1 + \pi_G \theta)(\rho + \delta) + 2\pi_G \beta \lambda + \pi_M \theta(\rho + \delta) + \pi_M \beta \lambda][\pi_M \theta(\rho + \delta) + \pi_M \beta \lambda]}{4\rho\alpha(\rho + \delta)^2}$$

By substituting $a_1$、$a_2$、$b_1$、$b_2$ to Formula (30), the optimal benefit function of the government and pharmaceutical enterprises is as follows:

$$V_G^{H*} = \frac{\pi_G \beta}{\rho + \delta} T + \frac{[2(\pi_1 + \pi_G \theta)(\rho + \delta) + 2\pi_G \beta \lambda + \pi_M \theta(\rho + \delta) + \pi_M \beta \lambda]^2}{8\rho\alpha(\rho + \delta)^2} \qquad (31)$$

$$V_M^{H*} = \frac{\pi_M \beta}{\rho + \delta} T + \frac{[2(\pi_1 + \pi_G \theta)(\rho + \delta) + 2\pi_G \beta \lambda + \pi_M \theta(\rho + \delta) + \pi_M \beta \lambda][\pi_M \theta(\rho + \delta) + \pi_M \beta \lambda]}{4\rho\alpha(\rho + \delta)^2} \qquad (32)$$

By substituting the derivative of Eq (31) and (32) with respect to T into Eq (25) and (27), the optimal technological innovation level k of pharmaceutical enterprises and the optimal subsidy coefficient F of the government can be obtained as Eq (22) and Eq (23) respectively.

Substituting Eq (22) into the equation of state gives Eq (21), i.e

$$T(t)^H = (T_0 + \frac{Z^H}{Y^H})e^{Y^H t} - \frac{Z^H}{Y^H}$$

Where $Y^H = -\delta$、$Z^H = \frac{2\lambda[(\pi_1 + \pi_G \theta)(\rho+\delta) + \pi_G \beta \lambda] + \lambda[\pi_M \theta(\rho+\delta) + \pi_M \beta \lambda]}{2\alpha(\rho+\delta)}$ Theorem 2 has been proved.

**Inference 2** According to Theorem 2, the partial derivation of various coefficients of the technological innovation level k of pharmaceutical enterprises shows that the technological innovation level of pharmaceutical enterprises is directly proportional to $\pi_1$ and $\pi_G$, and inversely proportional to $\pi_M$. In other words, the increase of technological innovation level of pharmaceutical enterprises can increase the social and economic benefits of the government. However, increasing the level of technological innovation means increasing the cost of technological innovation, so the economic benefits of enterprises decline. This is because technological innovation can increase consumer demand, expand market share, and thus reap certain profits. However, with the increase of technological innovation costs, the economic benefits brought by the expansion of the market are far from enough to offset the cost of enterprise losses, so the economic benefits of enterprises are reduced. In order to achieve better innovation income, enterprises need to determine an optimal level of technological innovation.

**Corollary 3** The financial subsidy F invested by the government is directly proportional to π1 and $\pi_G$, and inversely proportional to $\pi_M$, which means that when the government gives a certain financial subsidy to pharmaceutical enterprises, the social benefit and marginal income of the government will gradually increase with the increase of the subsidy proportion, while the marginal income of pharmaceutical enterprises will gradually decrease.

**Inference 4** Further compares the technological innovation level $k^N$ and $k^H$ of pharmaceutical enterprises under the two conditions of whether the government provides subsidies, and it can be known that government subsidies can help improve the technological innovation level of enterprises.

## 4.3 Contract of cooperation and coordination(C)

This section discusses the cooperation between the government and enterprises. The decision-makers of the system aim to maximize the benefits of the entire technological innovation system and determine the optimal level of technological innovation so as to achieve the optimal state of the entire system. Although this is difficult to achieve in the actual operation process, it is an idealized state, but it can be used as a reference to realize the Pareto improvement of the contract result. Here the superscript C represents the contract of cooperation and coordination.

**Theorem 3** Under cooperative coordination contract, the game equilibrium strategy of the system is

$$T(t)^C = (T_0 + \frac{Z^C}{Y^C})e^{Y^C t} - \frac{Z^C}{Y^C} \tag{33}$$

Where $Y^C = -\delta$, $Z^C = \frac{[\pi_1 + (\pi_G + \pi_M)\theta](\rho+\delta)\lambda + (\pi_G\beta + \pi_M\beta)\lambda^2}{\alpha(\rho+\delta)}$

$$k_S^C = \frac{[\pi_1 + (\pi_G + \pi_M)\theta](\rho+\delta) + (\pi_G\beta + \pi_M\beta)\lambda}{\alpha(\rho+\delta)} \tag{34}$$

It is proved that when the two cooperate, the technological innovation level of the enterprise is determined jointly by maximizing the benefits of the whole system, so the objective function of the system as follows:

$$\max_k J_S^C = \int_0^\infty e^{-\rho t}[\pi_1 k + (\pi_G + \pi_M)Q - C]dt \tag{35}$$

The optimal benefit function satisfies the HJB equation as follows:

$$\rho V_S^C = \max\{\pi_1 k + (\pi_G + \pi_M)Q - C + V_S^{C'}(\lambda k - \delta T)\} \tag{36}$$

Taking the first derivative of the above formula with respect to k yields:

$$k_S^C = \frac{\pi_1 + (\pi_G + \pi_M)\theta + V_S^{C'}\lambda}{\alpha} \tag{37}$$

It can be obtained by substituting it into the above HJB equation and simplifying it

$$\rho V_S^C = (\pi_G\beta + \pi_M\beta - V_S^{C'}\delta)T + (\pi_1 + \pi_M\theta + \pi_G\theta + V_S^{C'}\lambda)\frac{\pi_1 + (\pi_G + \pi_M)\theta + V_S^{C'}\lambda}{\alpha}$$
$$- \frac{[\pi_1 + (\pi_G + \pi_M)\theta + V_S^{C'}\lambda]^2}{2\alpha} \tag{38}$$

Similarly, the linear optimal benefit function for T is the solution to the HJB equation, assuming:

$$V_S^C(T) = n_1 T + n_2 \tag{39}$$

Then the parameters of the optimal function can be obtained by analogy with the coefficients of similar terms as follows:

$n_1 = \frac{\pi_G \beta + \pi_M \beta}{\rho + \delta}$, $n_2 = \frac{[(\pi_1 + \pi_M \theta + \pi_G \theta)(\rho + \delta) + \pi_G \beta \lambda + \pi_M \beta \lambda]^2}{\alpha \rho (\rho + \delta)^2}$ Substituting the above two parameters into the optimal benefit function can be obtained:

$$V_S^{C*} = \frac{\pi_G \beta + \pi_M \beta}{\rho + \delta} T + \frac{[(\pi_1 + \pi_M \theta + \pi_G \theta)(\rho + \delta) + \pi_G \beta \lambda + \pi_M \beta \lambda]^2}{\alpha \rho (\rho + \delta)^2} \tag{40}$$

By derivation of the above formula about T and substituting it into it, the optimal technological innovation level of pharmaceutical enterprises can be obtained as follows:

$$k_S^C = \frac{[\pi_1 + (\pi_G + \pi_M)\theta](\rho + \delta) + (\pi_G \beta + \pi_M \beta)\lambda}{\alpha(\rho + \delta)}$$

By substituting the above formula into the equation of state, the optimal trajectory of reputation can be obtained as

$$T(t)^C = (T_0 + \frac{Z^C}{Y^C}) e^{Y^C t} - \frac{Z^C}{Y^C}$$

Where $Y^C = -\delta$、$Z^C = \frac{[\pi_1 + (\pi_G + \pi_M)\theta](\rho + \delta)\lambda + (\pi_G \beta + \pi_M \beta)\lambda^2}{\alpha(\rho + \delta)}$ Theorem 3 is proved.

**Inference 5** In the coordination contract model, the technological innovation level k of pharmaceutical enterprises and the overall benefit $V_S^C$ of the system are positively correlated with $\pi_1$, $\pi_G$ and $\pi_M$. In other words, the higher the marginal income of pharmaceutical companies and the government, the higher the level of technological innovation of enterprises. It shows that the increase of the marginal income of the government can increase the technological innovation subsidy to enterprises, stimulate the technological innovation of enterprises, and increase the overall benefit of the system. The government should help enterprises share a certain proportion of innovation costs through reasonable investment to encourage pharmaceutical companies to be more proactive in technological innovation.

## 5 Comparative analysis

Through comparative analysis of the optimal technological innovation level k, corporate reputation and the optimal benefits of pharmaceutical enterprises in three different situations, the following conclusions can be drawn:

**Theorem 4** The technological innovation level of cooperative coordination contract is the best, $k_S^C > k^H > k^N$, that is, the technological innovation level of enterprises under cooperative contract is the largest, and the technological innovation level of enterprises without government subsidies is the smallest.

It is proved by the difference method, obtained by Eq (7), Eq (20),

$$k^H - k^N = \frac{2[(\pi_1 + \pi_G \theta)(\rho + \delta) + \pi_G \beta \lambda] + [\pi_M \theta(\rho + \delta) + \pi_M \beta \lambda]}{2\alpha(\rho + \delta)} - \frac{\pi_M \theta(\rho + \delta) + \pi_M \beta \lambda}{\alpha(\rho + \delta)} = \frac{2A - B}{2\alpha}$$

When 2A>B, there is, similarly using the difference method, obtained from Eq (20), Eq (31)

$$k^C - k^H = \frac{[\pi_1 + (\pi_G + \pi_M)\theta](\rho + \delta) + (\pi_G \beta + \pi_M \beta)\lambda}{\alpha(\rho + \delta)} - \frac{2[(\pi_1 + \pi_G \theta)(\rho + \delta) + \pi_G \beta \lambda] + [\pi_M \theta(\rho + \delta) + \pi_M \beta \lambda]}{2\alpha(\rho + \delta)} = \frac{B}{2\alpha} > 0$$

Therefore, when 2A>B, $k_S^C > k^H > k^N$

According to Theorem 4, when 2A>B, the enterprise innovation level under the cooperative coordination contract is the highest, indicating that when the government and pharmaceutical enterprises make coordinated decisions with the goal of maximizing the overall

benefits of the system, the enterprise's technological innovation level is the best, and the government's incentive effect on the enterprise is the best. Moreover, under non-cooperative contracts, the level of technological innovation increases with government subsidies. It can be seen that government subsidies have a certain threshold to stimulate the level of technological innovation of enterprises.

**Theorem 5** Corporate reputation is highest when cooperative contracts are coordinated and lowest when non-cooperative contracts are subsidized by the government.

Proof is obtained by difference of the above formula:

$T(t)^C - T(t)^H > 0$、 $T(t)^H - T(t)^N > 0$, therefore, $T(t)^C > T(t)^H > T(t)^N$ According to Theorem 5, government subsidies share the innovation cost of enterprises, thus helping enterprises realize technological innovation. Consumers who benefit from the drugs produced by enterprises through innovation give enterprises better evaluation and feedback, so enterprises gain a higher reputation. In addition, it is shown that government subsidies convey positive financial information about firms, indicate support for their innovative behaviour and contribute to their reputation. In addition, it is shown that government subsidies convey positive financial information about firms, indicate support for their innovative behavior and contribute to their reputation.

**Theorem 6** In the case of government subsidies, both the government and pharmaceutical enterprises achieve Pareto improvement in efficiency.

For the government, Formula (16) and Formula (27) obtain:

$$V_G^{H*} - V_G^{N*} = \frac{\pi_G \beta}{\rho + \delta} T + \frac{[2(\pi_1 + \pi_G \theta)(\rho + \delta) + 2\pi_G \beta \lambda) + \pi_M \theta(\rho + \delta) + \pi_M \beta \lambda]^2}{8\rho\alpha(\rho + \delta)^2} - \frac{\pi_G \beta}{\rho + \delta} T - \frac{[\pi_M \theta(\rho + \delta) + \pi_M \beta \lambda][(\pi_1 + \pi_G \theta)(\rho + \delta) + \pi_G \beta \lambda]}{\alpha\rho(\rho + \delta)^2}$$

$$= \frac{(2A - B)^2}{8\rho\alpha}$$

When 2A>B, $V_G^{H*} - V_G^{N*} > 0$, $V_G^{H*} > V_G^{N*}$

For enterprises, Eq (17) and Eq (28) can be similarly obtained:

$$V_M^{H*} - V_M^{N*} = \frac{\pi_M \beta}{\rho + \delta} T + \frac{[2(\pi_1 + \pi_G \theta)(\rho + \delta) + 2\pi_G \beta \lambda + \pi_M \theta(\rho + \delta) + \pi_M \beta \lambda][\pi_M \theta(\rho + \delta) + \pi_M \beta \lambda]}{4\rho\alpha(\rho + \delta)^2} - \frac{\pi_M \beta}{\rho + \delta} T - \frac{[\pi_M \beta \lambda + \pi_M \theta(\rho + \delta)]^2}{2\alpha\rho(\rho + \delta)^2}$$

$$= \frac{(2A - B)B}{4\rho\alpha}$$

When 2A>B, $V_M^{H*} - V_M^{N*} > 0$, $V_M^{H*} > V_M^{N*}$ According to Theorem 6, when 2A>B, in the case of government subsidies, the benefits of both the government and pharmaceutical enterprises are greater than the benefits of non-government subsidies, indicating that the government shares the cost of innovation of pharmaceutical enterprises by giving a certain proportion of financial subsidies, improved innovation performance of firms as well as social welfare, and realizes the Pareto improvement of benefits of both the government and enterprises.

**Inference 6** The Pareto improvement effect of both government and enterprise benefits under government subsidies is inversely proportional to the cost coefficient α of technological innovation efforts of pharmaceutical enterprises, and is directly proportional to the influence coefficient $\pi_1$ of technological innovation on social benefits and the difference of marginal income of government and enterprise.

**Theorem 7** The optimal benefit of the system under cooperative coordination contract is greater than that under the other two conditions.

$$V_G^H + V_M^H > V_G^N + V_M^N$$

From Eq (37), Eq (27) and Eq (28),

$$V_S^{C*} - (V_G^{H*} + V_M^{H*}) = \frac{4(A+B)^2 + B^2}{8\alpha\rho} > 0$$

Therefore, $V_S^{C*} > V_G^H + V_M^H > V_G^N + V_M^N$ According to theorem 6 and theorem 7, when 2A>B, compared with anarchic subsidy, government subsidy can realize the Pareto improvement of both government and enterprise in the system, and the optimal benefit of the whole system under the contract of government subsidy is greater than that of anarchic subsidy. The optimal benefit of the system under the cooperative coordination contract is greater than that under the government subsidy. It is worth noting that only when the respective benefits under the cooperative coordination contract between the government and the enterprise are greater than the non-coordinated cooperation can the optimal strategy of respective benefits under the government subsidy be accepted by both parties. The share of government and business in the system is determined by the negotiating power of the parties. Under the co-operation contract, the communication between the government and enterprises is relatively transparent, which greatly reduces the risk problems arising from information asymmetry, and the effectiveness of the system is greatly enhanced.

## 6 Numerical analysis

In order to better observe the effects of different government subsidy strategies and corporate reputation on firms' innovation level, this section uses Matlab for numerical simulation, and the basic parameter values are set as follows based on the practice of Xu Chunqiu [42], Chen Jingquan [49], Zhao Liming [50], et al:

$$\pi_1 = 1、\ \pi_G = 1、\ \pi_M = 1.5、\ \lambda = 2、\ \alpha = 6、\ \rho = 0.2、\ \delta = 0.1、\ \theta = 1、\ \beta = 0.5、\ \mathsf{T}_0 = 10$$

Firstly, based on the set benchmark parameter values, the relevant parameter values are substituted into the model. Fig 2 shows the comparative relationship between the benefits of the government and the enterprise under the non-cooperative contract when there is no government subsidy and when there is a government subsidy; Fig 3 shows the effect of parameter α on the Pareto improvement of the benefits of both government and enterprises under the non-cooperative contract of government subsidies.; Fig 4 shows the effect of parameter π1 on the coefficient of government subsidy F; Fig 5 reflects the effect of parameter λ on the innovation level of the enterprise K; Fig 6 shows the change of reputation of the pharmaceutical enterprise under the three kinds of contract over time; Fig 7 shows the comparison of the overall benefits of the system under the three contracts.

According to the above figures, observing the trend of the figure line, we can see that government subsidies and corporate reputation have a significant impact on the level of technological innovation of enterprises, which shows that government subsidies, corporate reputation and technological innovation have a significant correlation.

1. As can be seen from Fig 2, the use of government subsidy pact can realize the pareto improvement of the benefits of the government and pharmaceutical enterprises, and the improvement effect on the benefits of pharmaceutical enterprises is better than the improvement effect on the benefits of the government. Pharmaceutical companies are innovative R & D-oriented enterprises, the cost of innovation is large and high risk, due to a certain percentage of financial subsidies given to pharmaceutical companies, sharing part of the cost of innovation [23], while better incentives for pharmaceutical companies to technological innovation, research and development of new drugs to improve the demand for

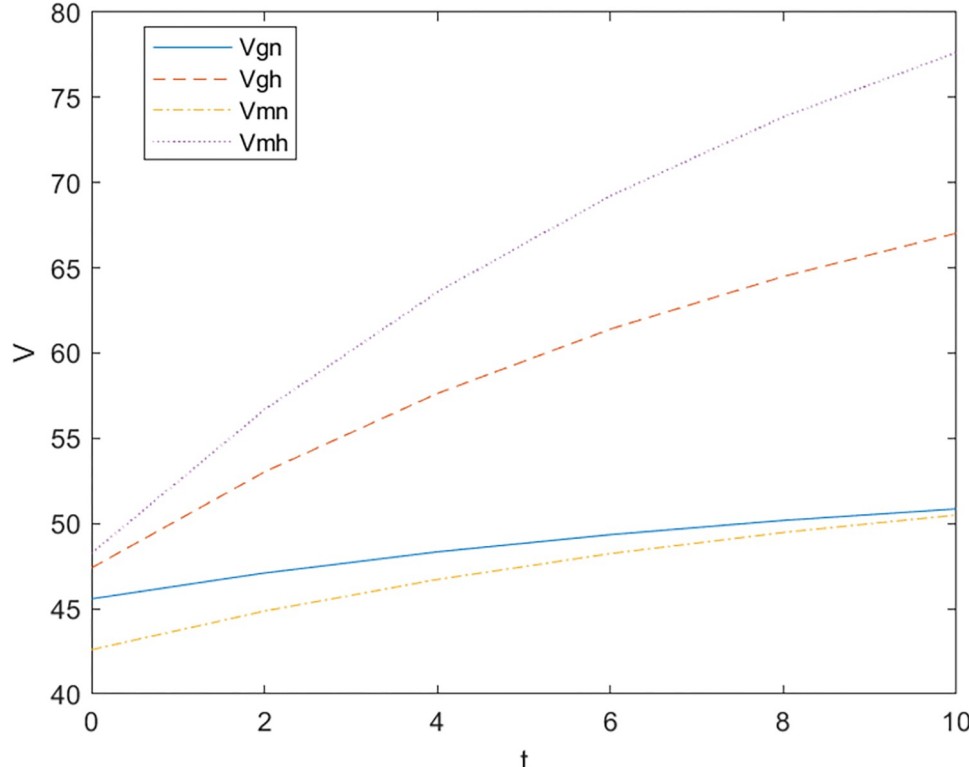

**Fig 2. Comparison chart of government-enterprise benefit relationship under government subsidy non-cooperative contract.**

medicines for patients, which brings an increase in the sales volume will indirectly enhance the benefits of the enterprise and the government, so that both sides of the benefits of the Pareto can be achieved to improve.

2. As can be seen in Fig 3, the Pareto improvement effect of government and enterprise benefits under the non-cooperative contract of government subsidies decreases with the increase of the coefficient of technological innovation costs of pharmaceutical companies. At this time, the government's benefit is lower than the enterprise's benefit. This is because under the non-cooperative contract, there is information asymmetry between the government and the enterprise, the government can not fully grasp the level of innovation efforts of the enterprise (the enterprise's innovation input), in the case of a certain amount of government subsidies, with the increase in the cost of innovation of the enterprise, the enterprise and the government's respective overall benefits are reduced with the increase of the cost of innovation, at this time the worse the effect of the government's subsidy incentives, so the government subsidy decision-making can not be a set in stone. Therefore, when making decisions on subsidies, the government should not be static, but should consider as much as possible the cost pressure of pharmaceutical enterprises in technological innovation, and give appropriate subsidies based on the actual situation of enterprise innovation, in addition to more channels of support, such as shortening the cost of the time to market new drugs, and giving more protection to the patent of the new drug R & D, and so on.

3. It can be seen from Fig 4 that as the parameter $\pi 1$ increases, that is, the impact of corporate technological innovation investment on government social benefits increases, the subsidy

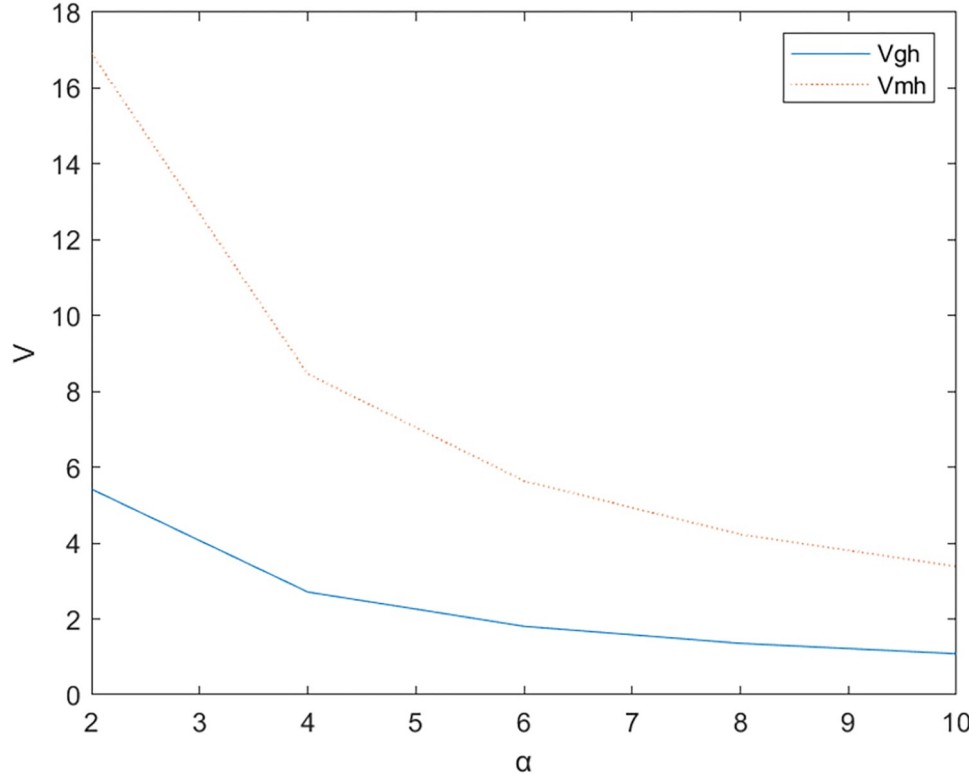

**Fig 3. The influence of parameter α on the improvement effect of government subsidy non-cooperative contract pareto.**

coefficient given by the government is also greater, and the growth rate of the subsidy coefficient slows down. It explains that the more sensitive consumers are to innovative drugs, the greater the social benefits brought by innovation. $\pi 1$ can indicate the degree of patient demand for new drugs and special drugs. When $\pi 1$ is large, patients have a greater demand for scarce drugs. The government has higher expectations and attention for the innovation and R&D investment of pharmaceutical companies. In order to stimulate the innovation and R&D of pharmaceutical companies and meet the needs of some patients, especially rare patients, the government often chooses to give a large proportion of financial subsidies. For example, in the new corona period, in order to encourage pharmaceutical companies to develop new corona vaccines. The government will give pharmaceutical companies greater financial subsidies (cash incentives, tax incentives, etc.) to support innovative research and development. When $\pi 1$ is small, that is, some specific drugs on the market are relatively saturated, and the patient's demand for new drugs is low, the government has no higher expectation for the technological innovation of pharmaceutical companies, and will not give a higher subsidy coefficient. At this time, in order to seek government subsidies, enterprises need to actively respond to consumers' medication needs, and continuously innovate and improve product quality based on patient feedback.

4. Fig 5 indicates that the larger the coefficient λ of the influence of innovation level on corporate reputation, the higher the level of innovation effort of the enterprise, and when the government gives subsidies, the more sensitive the corporate reputation is to the coefficient of innovation effort, and the level of innovation of the enterprise is more affected by the level of reputation. Therefore, the government gives subsidies to enterprises, which will increase

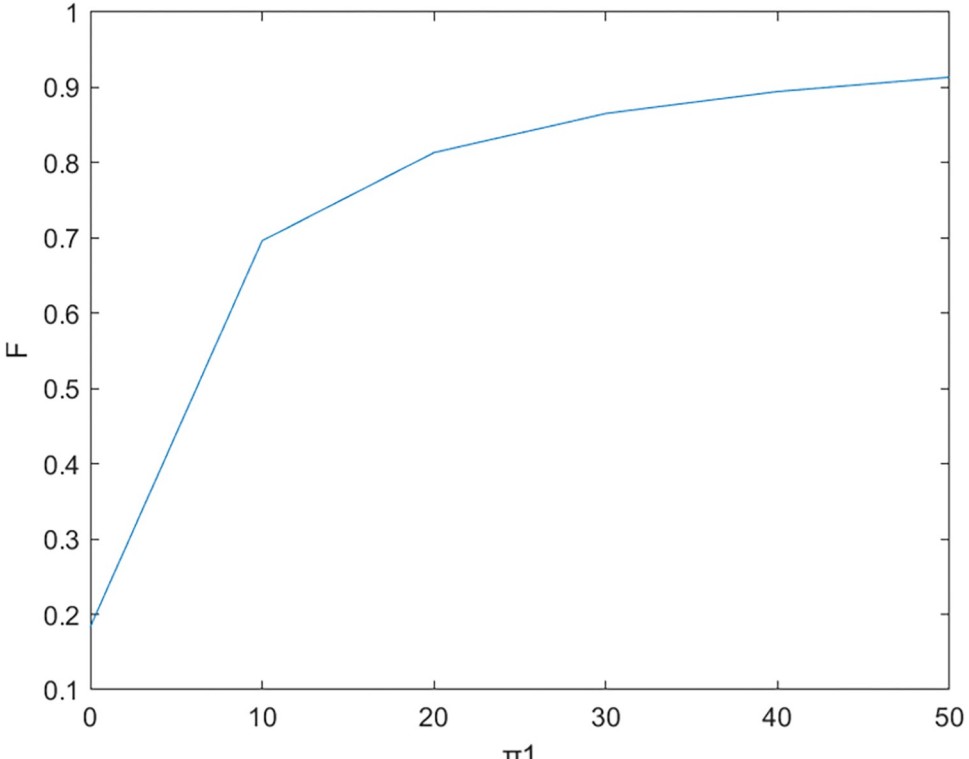

**Fig 4. The influence of parameter π1 on government subsidy coefficient.**

the sensitivity of corporate reputation to the coefficient of innovation effort, and pharmaceutical companies should attract more consumers by increasing the publicity of innovation, increasing the investment in innovation, and so on, which improves the reputation and also improves the level of innovation effort.

5. As can be seen from Fig 6, the reputation of enterprises under the cooperative coordination contract is the highest, and the reputation of enterprises under the non-cooperative contract without government subsidies is the lowest, which on the one hand indicates that government subsidies promote their technological innovations and enterprise innovations further enhance the level of enterprise reputations. On the other hand, government subsidies have an information transfer effect [21, 54, 55], which provides publicity for firms' innovative activities and thus enhances their reputation. And with the increase of time, the enterprise's reputation has an upward trend.

6. As can be seen from Fig 7, the system benefits of the cooperative coordination contract are the highest in the three cases and the system benefits are the lowest in the case of no government subsidy, and the government subsidy can realize the Pareto improvement of the overall system benefits. In addition, the overall system benefit under the cooperative coordination contract is much larger than the system benefit under the two scenarios under the non-cooperative contract, which verifies the previous inference and indicates that the decision-making during the cooperative coordination contract is better than the decision-making under the non-cooperative contract. Combined with Figs 5 and 6, it can be found that with the increase of time, the improvement of enterprise reputation eventually brings the increase of benefits, which is because a good reputation can attract more investment

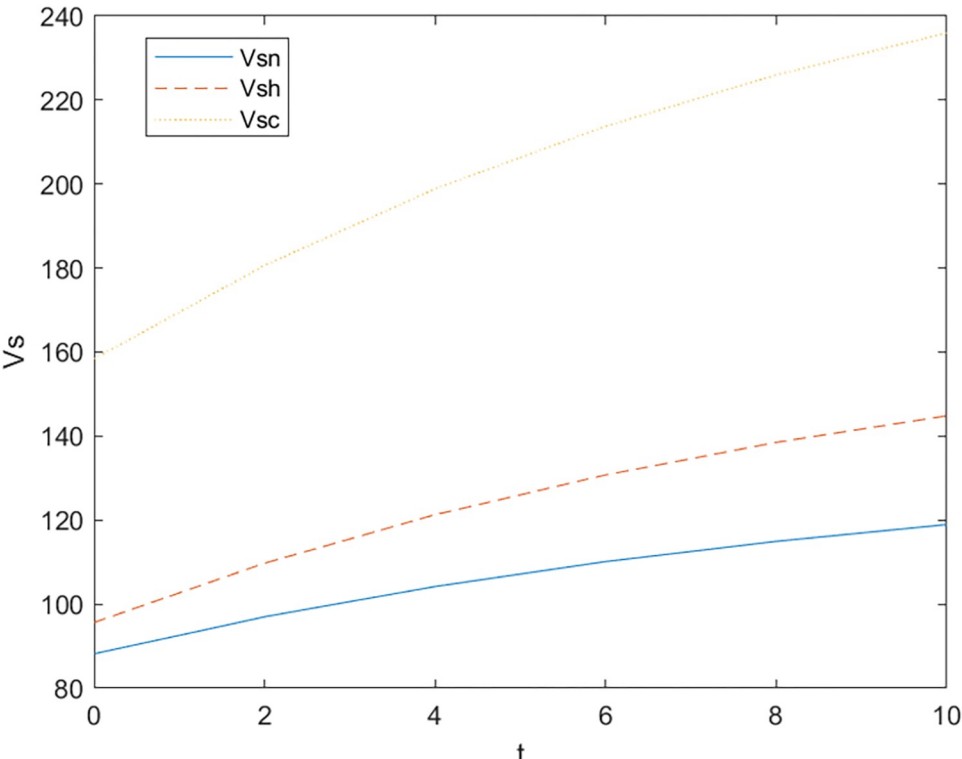

**Fig 5. The influence of parameter λ on the level of innovation under different subsidies.**

and customers' favor, provide economic support and prime mover for enterprise innovation, and help the enterprise to obtain a greater competitive advantage, and ultimately realize the growth of profits.

## 7 Discussion

The technological innovation behavior of pharmaceutical enterprises demonstrates the enterprise's sense of social responsibility, which is conducive to enhancing the image and brand reputation of the enterprise. However, the innovative research and development of pharmaceutical enterprises has always been a time-consuming and laborious project. To accomplish this difficult and complex task, it is difficult to realize only relying on the strength of pharmaceutical companies themselves. The power of the government becomes very important. In this paper, the optimal decisions of pharmaceutical enterprises and the government under different subsidy strategies are compared and analyzed by constructing a differential game model, and the conclusions of the study are as follows:

1. When there is no government subsidy, the introduction of the non-cooperative contract of government subsidy, when 2A>B, the level of innovation efforts of pharmaceutical enterprises, the level of reputation, and the profits of both the government and the enterprise are improved, that is to say, the Pareto improvement of the government benefit, the enterprise benefit, and the overall benefit is realised. Therefore, the introduction of the non-cooperative contract with government subsidies not only satisfies the patients' demand for medicines, but also improves the profits of pharmaceutical enterprises, thus enhancing the core competitiveness of the enterprises; it can explain the motivation of the pharmaceutical

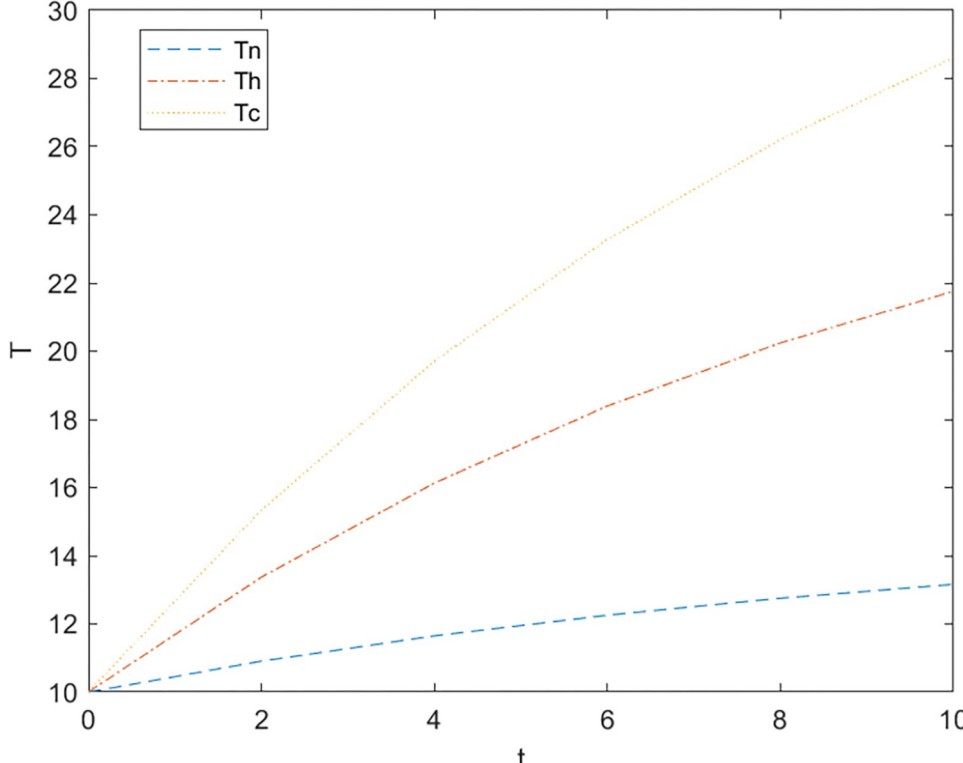

**Fig 6. The change of corporate reputation over time in three cases.**

enterprises to choose high-risk innovation and R&D, and it also verifies the reasonableness of the government's policy of subsidies for the pharmaceutical enterprises that carry out innovative research and development.

2. The more sensitive the enterprise's reputation is to the coefficient of technological innovation, the more it can improve the level of enterprise innovation. The implementation of technological innovation by enterprises belongs to the implicit type of social responsibility behaviour, therefore, the extent to which the technological innovation of enterprises can affect the improvement of corporate reputation, in turn, has a different impact on the level of corporate innovation efforts. It also depends on the degree of consumer demand for innovative medicines. Firms should continue to innovate in order to maintain a certain level of reputation level.

3. Under the government-enterprise co-operation pact, the level of innovation and reputation of firms is optimal, the overall efficiency of the system is maximised, and the efficiency of the government's innovation subsidy is significantly improved.

## 7.1 Management insights and recommendations

In order to improve the efficiency of government subsidies and the high-level innovation ability of enterprises, the following suggestions are put forward:

1. The government in order to incentivise pharmaceutical enterprises to innovate, to achieve their own social and economic benefits of maximization, should give pharmaceutical enterprises appropriate financial subsidies, the amount of government subsidies should not be

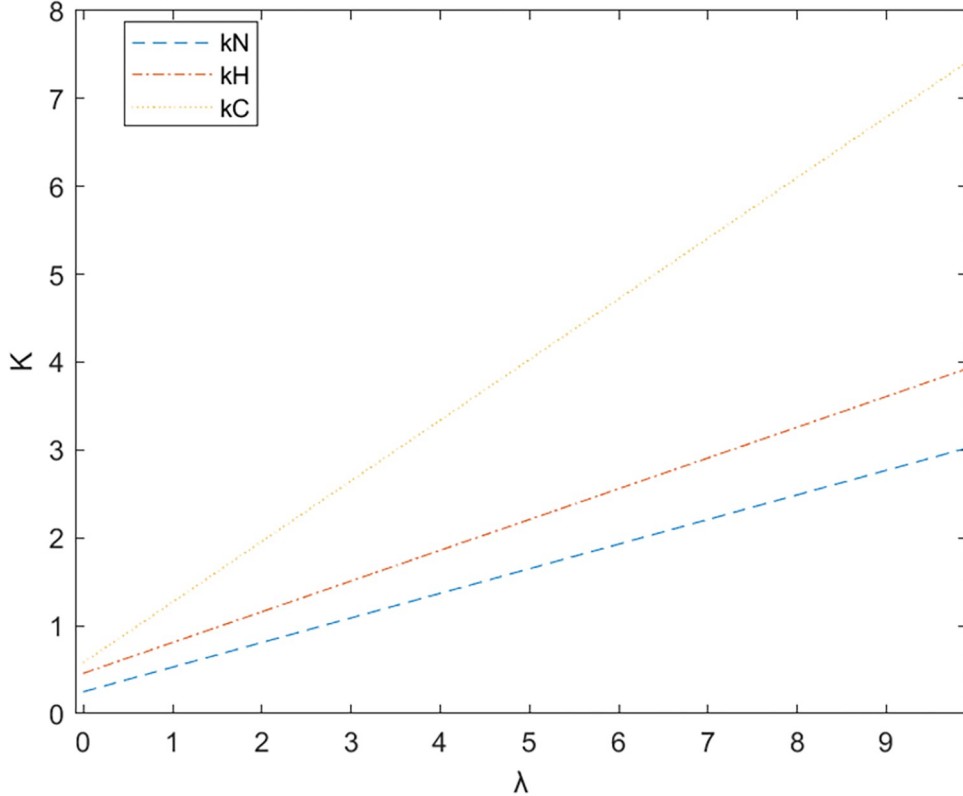

**Fig 7. Comparison of system benefits under three contracts.**

static, nor the more the better, but within a certain range with the changes in the cost of technology of the enterprise and the dynamic adjustment of the. At the same time, in order to enhance the efficiency of government subsidies, the government should improve the subsidy mechanism and innovative appraisal system to improve the transparency of government subsidy information, to convey the information of good economic behaviour of enterprises. It should also expand regulatory channels and make use of the relatively objective factor of corporate reputation brought about by feedback on consumer behaviour to effectively grasp the degree of innovation of the supporting enterprises, thus reducing the risk of information asymmetry.

2. Pharmaceutical companies should actively respond to the government's subsidy policy, the use of government subsidies to actively invest in innovation and research and development, improve the quality of medicines, and constantly explore and meet consumer demand for medicines, consumer demand-oriented, the higher the degree of consumer demand for medicines, the more the government pays attention to corporate innovation, the greater the coefficient of subsidies given. In addition, pharmaceutical companies should actively fulfill the corporate social responsibility based on technological innovation, give full play to the advertising effect of social responsibility to improve the reputation, and reduce the information asymmetry between enterprises and stakeholders. It is necessary to maintain a good level of reputation through continuous innovation, once the enterprise stops innovating, the corporate reputation will then decay, and the decline in reputation will in turn inhibit corporate innovation. Therefore, pharmaceutical companies can only form a good sustainable development through continuous innovation.

3. Pharmaceutical companies and the government can reach cooperation, the formation of synergies, reduce the risk of information asymmetry and the obstacles of unnecessary communication links, and smooth the exchange of procedures. The government for the realization of social benefits and their own economic benefits to enterprises a certain percentage of innovation subsidies, pharmaceutical companies in order to achieve their own development, carry out technological innovation and create more own wealth and social welfare, which in turn increases the overall benefits of the government, the formation of a virtuous circle, and ultimately the realization of the entire pharmaceutical industry and even the development of society.

## 7.2 Theoretical and practical implications

The theoretical model established in this paper can be used as a reference to study the reputation effect of technological innovation. Most of the previous studies on the reputation effect of technological innovation have been tested through empirical studies, so subsequent studies can be guided by this. In addition, traditional research on government subsidies tends to treat government subsidies as the most exogenous variable, and our study analyzes the government's optimal subsidy decision under information asymmetry from the government's perspective. Based on this work subsequent studies can further combine the classical scenarios in supply chain systems with government subsidies and extend to more diverse subsidy scenarios. Finally, while previous studies tend to study the impact of government subsidies on technological innovation decisions from a single perspective, this paper innovatively incorporates corporate reputation and government subsidies into the research field of technological innovation decisions and explores the mediating mechanism of corporate reputation between government subsidies and technological innovation from a new perspective.

This study also has some practical guidance:

1. The three contract models have clear theoretical solutions, which to a certain extent solve the problem of difficult practical implementation. Based on the clear theoretical solutions of the model, the government can optimize the model parameters and decision-making contract types according to its political needs and its own economic development. Similarly, pharmaceutical companies can develop more precise response strategies according to their own status quo and innovation needs.

2. The contractual model has a signalling function. Generally speaking, government subsidies will convey the government's support for the enterprise, which is conducive to conveying the signal to the public that the enterprise actively fulfils its social responsibility for technological innovation, thus attracting more social support. After model optimisation and parameter assignment analysis, the government can roughly judge the degree of effort of pharmaceutical enterprises' technological innovation, and can also remind pharmaceutical enterprises not to speculate and fake innovation.

3. This study helps the government to build a desirable social and economic order. Through the assignment and analysis of the parameters of the reputation model, the government can not only formulate appropriate subsidy coefficients and improve the efficiency of subsidies, but also help to guide and optimise the content, process and publicity form of corporate social responsibility. At the same time, the law of change of reputation also reflects to a certain extent the degree of attention and demand of patients for innovative drugs. Pharmaceutical companies can make appropriate innovation decisions based on the pattern of change in reputation, and find innovative ways from the needs of patients.

## 7.3 Limitations and perspectives

There are many shortcomings in this paper, which also provide directions for future research. Firstly, this paper does not consider the uncertainty of demand, assuming that consumers with feedback function are sensitive people with certain demand for innovative drugs, such as patients with rare diseases, etc. However, most consumers in real life are not sensitive to innovative drugs, and this assumption should be further relaxed and expanded in the future. Second, the impact of different subsidy methods on the level of technological innovation of enterprises may be different. Although this paper considers the impact of the amount of government subsidies on enterprise innovation, it does not consider the impact of other forms of government subsidies (subsidy time, subsidy mode, etc.) on enterprise technological innovation, and the impact of different types of government subsidies on various aspects of enterprise innovation (R&D, production, etc.) can be further studied in the future. Finally, the relevant parameters are simplified in order to obtain the analytical solution of the game problem, and seeking numerical solutions of the differential equations for non-degenerate problems will be an important direction in the subsequent research.

## Acknowledgments

Thanks to Professor Zhe Huang for her careful guidance and help on my thesis. Thanks for the passionate guidance and help of fellow students Fan Zhang and senior fellow apprentice Dr. Xiangqi Zhao.

## Author Contributions

**Formal analysis:** Yu Kang.

**Supervision:** Zhe Huang.

**Writing – original draft:** Yu Kang.

**Writing – review & editing:** Yu Kang.

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
