## [Decision Letter · Decision Letter 0]

17 Oct 2023

PONE-D-23-30152Research on technological innovation decision-making considering government subsidies and corporate reputationPLOS ONE

Dear Dr. Huang,

Thank you for submitting your manuscript to PLOS ONE. After careful consideration, we feel that it has merit but does not fully meet PLOS ONE’s publication criteria as it currently stands. Therefore, we invite you to submit a revised version of the manuscript that addresses the points raised during the review process.

We look forward to receiving your revised manuscript.

Kind regards,

Jitendra Yadav, Ph.D.

Academic Editor

PLOS ONE

Journal Requirements:

"No" 

Reviewers' comments:

Reviewer's Responses to Questions

**Comments to the Author**

1. Is the manuscript technically sound, and do the data support the conclusions?

Reviewer #1: Partly

Reviewer #2: Yes

2. Has the statistical analysis been performed appropriately and rigorously? 

Reviewer #1: Yes

Reviewer #2: I Don't Know

3. Have the authors made all data underlying the findings in their manuscript fully available?

Reviewer #1: No

Reviewer #2: Yes

4. Is the manuscript presented in an intelligible fashion and written in standard English?

Reviewer #1: No

Reviewer #2: Yes

5. Review Comments to the Author

Reviewer #1: While it is important to acknowledge the potential significance of research on technological innovation decision-making, especially in the context of government subsidies and corporate reputation, there are several critical issues and shortcomings that need to be addressed before this study can be considered for publication.

Reviewer #2: Dear Author(s),

I trust this message finds you well, I am writing to provide feedback on the paper titled “Research on technological innovation decision-making considering government subsidies and corporate reputation” submitted to PLOS ONE for consideration and I would like to offer some feedback aimed at strengthening your work further:

Strengthening CSR Alignment with Technological Innovation:

The paper mentions Corporate Social Responsibility (CSR) alignment with technological innovation, but the discussion in the manuscript could be more robust. To make your work more relevant and significant, delve deeper into how your model aligns with CSR principles, values, and practices. This alignment will underscore the paper's importance in the context of CSR-related research and corporate reputation.

Enhancing Practical Implications:

While your paper does address practical implications, there is room for a more comprehensive exploration of this aspect. Consider providing a clearer and more detailed outline of how your developed model can be practically applied. Expanding on practical implications will make your research more valuable to both practitioners and researchers in your field.

Structure

The structure of a research paper plays a vital role in conveying the research effectively to readers. To enhance the clarity and coherence of your work, I strongly recommend restructuring your paper to follow a more classical and conventional format commonly used in your field. A well-structured paper typically includes the following sections: introduction, literature review, methodology, discussion, conclusion, references.

I sincerely believe that addressing these aspects will enhance the quality of your work. Your research has the potential to make a meaningful contribution to your field, and I'm excited to see it progress.

Thank you for your dedication to advancing your research, and I look forward to the updated version of your paper.

6. PLOS authors have the option to publish the peer review history of their article (what does this mean?). If published, this will include your full peer review and any attached files.

Reviewer #1: No

Reviewer #2: No

---

## [Author Response · Author response to Decision Letter 0]

20 Nov 2023

First of all, I would like to express my sincere appreciation to you for taking the time to read and modify my article. Thank you for your valuable suggestions. You have made comprehensive corrections to the structure, content, research methods and results of my paper. It plays a very important role in improving the quality of my thesis. To the request of the journal, make the following response:

1、The authors have revised the manuscript according to the stylistic requirements of PLOS ONE

2、The authors have provided shared data code according to the guidelines

3、The author has re-provided the funding information in the cover letter

4、The author has updated the financial disclosure in the cover letter

5、The author has provided a public repository for the minimum data set in the cover letter

6、The author has added information about the data availability statement to the cover letter

For the constructive suggestions put forward by the reviewers, the author replies point by point, as follows:

Comment 1：An organized conceptual framework that may direct the investigation is missing from the text. It is not obvious how business repute and government subsidies are factored into the decision-making process. The study's theoretical underpinnings and a precise research strategy should be established by the writers.

Response: Thank you very much for the valuable suggestions of the reviewers ! We have improved it in the introduction of the article, and described in detail the process of corporate reputation and government subsidies being included in the decision-making. The modified part has been marked red in the introduction part.

Comment 2: The literature review is insufficient and does not sufficiently highlight the holes that are currently in the area. To determine the importance of their findings, the writers must perform a more thorough examination of the pertinent literature.

Response: Thanks for the valuable suggestions ! We re-combed and integrated the relevant literature, and gave the importance and innovation of this study. The above content has been shown in ' 2 Literature Review ' and has been marked red.

Comment 3: The approach section is not specific enough. It is unclear how the data was gathered, which factors were taken into account, and which analytical techniques were employed. The sample size and data sources are also not disclosed. For the study to be believed, a solid technique is required.

Response: Thank you very much for the valuable suggestions of the reviewers ! The data involved in the article is mainly the simulation part of the model. It is mainly through the assignment of key parameters to observe the changes of other variables in order to analyze the relevant trends.

Comment 4: It is difficult to assess the quality and validity of the data utilized in the study without knowledge of the data sources and the data gathering procedure. The data gathering process should be clearly described by the authors, along with steps taken to guarantee the quality and trustworthiness of the data.

Response: Thank you very much for the valuable suggestions of the reviewers ! Regarding the quality and effectiveness of the data, this paper refers to the processing methods of similar studies and assigns key parameters. The relevant references have been listed and marked red in the corresponding position of the ' 6 Numerical analysis ' section.

Comment 5: If a quantitative analysis was done, the authors should go into great detail about the statistical procedures and tests that were utilized. This outlines the regression models and significance tests used in comprehensive detail. It is hard to judge the analysis's rigor in the absence of these data.

Response: Thanks for the review expert's reminder! In this paper, the drawing software Matlab is used to analyze the analytical solutions obtained above. The reliability of the results is verified by analyzing the images. The specific modification has been specifically explained in the section of numerical analysis, and the relevant literature on the application of this method in similar studies is listed, including how to assign the key parameters and the range of the assignment. Corrections have been marked in red.

Comment 6: The research seems to show a causal relationship between business reputation, government subsidies, and choices about technological innovation. It is important to make it clear that correlation does not indicate causality. The research design's constraints and potential endogeneity problems should be discussed by the authors.

Response: Thanks for the guidance of reviewers ! This paper constructs a differential game model, and combines numerical simulation to confirm the impact of government subsidies and corporate reputation on technological innovation decision-making. It is summarized after Figure 6, and the relevant parts have been marked red.

Comment 7: The discussion section ought to include more in-depth explanations of the results' implications. What are the practical ramifications for firms and decision-makers, and how do government subsidies and company reputation impact decision-making in technological innovation? The study ought to outline a more direct route from research to practical applications.

Response: Thank the reviewers for their valuable suggestions ! We supplement the relevant content in the conclusion, and make a more in-depth explanation and description of the actual impact of the results on enterprises and governments, as well as how government subsidies and corporate reputation affect decision-making in technological innovation.

Comment 8: The authors' inclusion of specific recommendations for practitioners and policymakers based on their results would be advantageous. This would increase the research's applicability in real-world settings.

Response: Thanks to the reviewer 's kind reminder ! We have given relevant suggestions to the government and pharmaceutical companies, and reflected in the conclusion, the relevant content has been marked red.

Comment 9: The paper needs to be better organized overall. It is challenging to follow the argument's logical progression. To make the document more coherent and clear, it should be reorganized.

Response: Thank you for the professional advice of the reviewers ! We carefully review the writing context of the article, and give the logical arrangement of this article in the introduction. This part is reflected in the last paragraph of the introduction, which has been marked red.

Comment 10: The study's main conclusions, contributions, and limitations should be summed up in the conclusion. It ought to additionally emphasize the lines of inquiry for more study in this field.

Response: Thanks for your valuable suggestions ! We have re-combed the main conclusions, contributions and limitations of the article in the ' 7 conclusions ', and made a red mark.

Comment 11: To solve the conceptual, procedural, and analytical flaws, the study needs to be significantly revised. To be taken into consideration for publishing, the authors must present a more substantial and well-organized study that contribute to the body of knowledge on business reputation, government funding, and technological innovation decision-making.

Response: We sincerely appreciate the valuable comments! We re-examined the article and revised the comments of the reviewers in detail to make the article more logical and easy to read.

---

## [Decision Letter · Decision Letter 1]

15 Dec 2023

PONE-D-23-30152R1Research on technological innovation decision-making considering government subsidies and corporate reputationPLOS ONE

Dear Dr. Huang,

Thank you for submitting your manuscript to PLOS ONE. After careful consideration, we feel that it has merit but does not fully meet PLOS ONE’s publication criteria as it currently stands. Therefore, we invite you to submit a revised version of the manuscript that addresses the points raised during the review process.

We look forward to receiving your revised manuscript.

Kind regards,

Jitendra Yadav, Ph.D.

Academic Editor

PLOS ONE

Journal Requirements:

Additional Editor Comments:

The reviewers have provided valuable feedback that, if addressed appropriately, will enhance the overall quality and scholarly impact of your work. The primary concerns raised include:

1. Unclear Research Need: The manuscript lacks a clear articulation of the research need, making it challenging for readers to understand the significance and relevance of the study.

2. Unclear Research Gap: The research gap is not adequately addressed or defined, leaving readers without a clear understanding of the existing knowledge deficiencies that the study aims to address.

3. Missing Methodology Section: The absence of a well-defined methodology section hinders the ability to evaluate the robustness and reliability of the research design.

4. Lack of Description of Data Collection Methodology: The manuscript does not provide a detailed description of the data collection methodology, making it difficult to assess the validity and reliability of the gathered data.

5. Absence of Description of Analysis Procedure: The analysis procedure is not sufficiently explained, leaving readers and reviewers without a clear understanding of how the data were processed and interpreted.

6. Failure to Explain Contribution to Existing Knowledge and Practice: The manuscript lacks a compelling discussion on how the research contributes to extending the existing body of knowledge and practice in the field. It is essential to explicitly articulate the novelty and significance of the findings.

To guide you through the revision process, we have attached the detailed comments and recommendations from the reviewers. Please thoroughly address each point and clearly indicate the changes made in your revised manuscript.

Reviewers' comments:

Reviewer's Responses to Questions

**Comments to the Author**

1. If the authors have adequately addressed your comments raised in a previous round of review and you feel that this manuscript is now acceptable for publication, you may indicate that here to bypass the “Comments to the Author” section, enter your conflict of interest statement in the “Confidential to Editor” section, and submit your "Accept" recommendation.

Reviewer #1: (No Response)

Reviewer #3: (No Response)

2. Is the manuscript technically sound, and do the data support the conclusions?

Reviewer #1: Partly

Reviewer #3: No

3. Has the statistical analysis been performed appropriately and rigorously? 

Reviewer #1: Yes

Reviewer #3: No

4. Have the authors made all data underlying the findings in their manuscript fully available?

Reviewer #1: Yes

Reviewer #3: No

5. Is the manuscript presented in an intelligible fashion and written in standard English?

Reviewer #1: No

Reviewer #3: (No Response)

6. Review Comments to the Author

Reviewer #1: Remember to carefully address each reviewer's comment and make necessary revisions to enhance the quality and impact of your research paper.

Reviewer #3: (No Response)

7. PLOS authors have the option to publish the peer review history of their article (what does this mean?). If published, this will include your full peer review and any attached files.

Reviewer #1: No

Reviewer #3: **Yes: **Avikshit Yadav

---

## [Author Response · Author response to Decision Letter 1]

24 Jan 2024

Dear Editor and Reviewers,

We appreciate your professional comments on our article. As you are concerned, our article has several issues that need to be addressed. Based on your suggestions, we have made extensive changes to the previous manuscript! The reviewers ' comments are listed in italics below and numbered separately. Our reply is given in a normal font, and the specific revisions or additions to the manuscript are marked in red in the revised manuscript.

Response to reviewer 1 comments:

1.Consider how well your literature review lays the groundwork for your study and how thorough it is. Make sure you have included pertinent research on government subsidies, company reputation, and technical innovation.

We sincerely appreciated the valuable comments. We have checked the literature carefully and added more references on government subsidies, company reputation, and technical innovation into the Literature Review part in the revised manuscript.

2.An important component is interpreting the findings. Talk about how your findings affect the way that decisions about technological innovation are made, taking into account both business reputation and government subsidies. Emphasize how your study is applicable in the real world.

We sincerely appreciated the reviewers for their valuable suggestions! According to the results of the study, the Pareto improvement in system efficiency can be achieved under the government co-operative contract, when the level of technological innovation and the level of reputation of the firms are optimal compared to the non-cooperative contract without government subsidy and under government subsidy. The more sensitive the firm's reputation is to innovation efforts, the more it improves the firm's innovation level. In addition, the greater the social welfare brought about by the firm's innovation, the more it increases the government's subsidy coefficient. Therefore, enterprises should be oriented to consumer demand and maintain a good reputation level through continuous innovation, a good reputation level can attract government investment, government subsidies can share the innovation cost of enterprises, and under the government subsidy strategy, enterprises can achieve the improvement of the benefits of both government and enterprises through technological innovation. We attempted to use quantitative research methods to obtain the theoretical solutions of the three contract models, which to a certain extent solved the problem of practical implementation difficulties. Based on the explicit theoretical solution of this model, the government can optimise the model parameters and decide the type of contract according to the political demand and its own economic development status, etc. Similarly, the pharmaceutical enterprises can formulate a more accurate response strategy according to the current situation of the enterprise and the demand for innovation. In addition, we have tested the validity of the results through case studies based on the literature. More detailed modifications have been added to specific paragraphs in the text (7 Conclusion) and are marked in red.

3.Make sure that, in the context of technical innovation, your research successfully incorporates the two factors of government subsidies and business reputation. Talk about the interactions and effects these elements have on making decisions.

Thank you for your significant reminding. We carefully reviewed the article and revised and improved the relevant research details. In 6 Numerical analysis, we added the content of the impact of the coefficient of technological innovation on reputation λ on the level of technological innovation K. And the effects of government subsidies and corporate reputation on firms' innovation decisions, as well as their interactions, are also fully analysed. Specific changes have been marked in red.

4.Give a concise summary of your research's management and policy implications. What conclusions can you draw from your research, and how can politicians or business implement these conclusions in real-world settings?

Thank you for your constructive advice. A total of three main findings were obtained from our study and based on the findings:

（1）When there is no government subsidy, the introduction of the non-cooperative contract of government subsidy, when 2A>B, the level of innovation efforts of pharmaceutical enterprises, the level of reputation, and the profits of both the government and the enterprise are improved, that is to say, the Pareto improvement of the government benefit, the enterprise benefit, and the overall benefit is realised. 

（2）The more sensitive the enterprise's reputation is to the coefficient of technological innovation, the more it can improve the level of enterprise innovation.

（3）Under the government-enterprise co-operation pact, the level of innovation and reputation of firms is optimal, the overall efficiency of the system is maximised, and the efficiency of the government's innovation subsidy is significantly improved.

 Reference recommendations are made for policy makers in government and pharmaceutical companies. Specific changes to the conclusions and policy recommendations are detailed in the‘7 conclusion’ section of the article and are marked in red.

5.Clearly state how your study adds to the corpus of knowledge already in existence. How does your research contribute to our knowledge of how decisions about technological innovation are made?

Thank you for pointing this out. The innovativeness of this paper is mainly reflected in the following aspects: firstly, a series of conduction relationships are established through the dissemination of reputation. A firm's technological innovation efforts affect the firm's reputation → the firm's reputation affects consumer demand → the demand further affects the firm's profit. By modelling this conduction relationship, a mathematical description of the problem is formed. Second, from the government's standpoint, the advantages and disadvantages of various participation methods are examined in order to reduce information asymmetry and obtain optimal subsidy efficiency. Thirdly, from the perspective of the firms, appropriate technological innovation decisions are made based on the subsidy strategy set by the government in order to maintain a good corporate reputation and further attract government support to stimulate corporate innovation, creating a virtuous circle and achieving sustainable development. More detailed explanations are given in the ‘2 literature review’ and in the‘7.2 Theoretical and practical implications’ of the ‘7 conclusions ’ section of the text and are marked in red

6.Write a summary of your main conclusions and their wider ramifications. Summarize the importance of your findings and offer potential directions for further investigation.

The valuable suggestions of the reviewers are highly appreciated! We have given the main contribution of the paper in the penultimate natural paragraph of the‘2 Introduction’ as well as in the‘7 Conclusion’ section respectively adding the main conclusions of the article, theoretical and practical implications and further research work in the future, which are marked in red.

7.Make sure your essay has a logical framework, is well-written, and is clear. Be mindful of your spelling, grammar, and formatting.

Thank you for your kind reminding. In order to make the content of the article clearer and more logical, we carefully reviewed the content of all chapters and made the language of the article smoother and more standardised through repeated revisions.

Response to reviewer 3 comments:

1.Your abstract does not highlight the specifics of your research or findings but contains too much background information. Some details of your research would be nice, for example, numbers addressing the sample, data, percentage improvement, etc. An abstract with some details helps show the impact of your research. This does not mean making the abstract longer! Shorter is better; remove some of the background material and add some details of your research. Again, it is good to provide some specifics (e.g., sample size, dataset size, numbers from results, etc.)

We think this is an excellent suggestion. We have re-written this part according to the reviewer’s suggestion. The specific changes can be seen in the Abstract section of the text and are marked in red.

2.Avoid repetition in the manuscript (e.g., in the Abstract section, ‘………under three conditions: non-cooperative contract subsidized by the government, non-cooperative contract subsidized by the government, and cooperative coordination contract’).

Thanks for your helpful suggestion. We have scrutinised the article and deleted the corresponding repetitions.

3.Information about your dataset is missing. So, you need to explicitly provide a description of your dataset(s), in the manuscript, which is currently lacking. If the author(s) has provided only simulation part of the model, how the simulation part will be able to address the real world problem!

Thanks for the kind reminders from the reviewing experts! In this paper, we construct a differential game model and solve it by using Hamilton-Jacobi equation, and in‘6 Numerical analysis’, we refer to the research results of the related literature as well as the real situation to set the parameter values in the model of this paper in a reasonable way. In fact, simulation can help us predict and model the behaviour and results in the real world, and by simulating the actual situation, we can better understand and solve the real problems. Although simulation cannot completely replace the reality, it can help us better understand the real problems and provide valuable references for the solution of real problems through parameter adjustment and model optimisation. There are a lot of scholars who use simulation for case analysis to solve real problems. These cases can prove the practicality and effectiveness of simulation. We list the relevant references in the text and mark them in red.

4.The author(s) must more clearly highlight their research’s theoretical and practical implications; it is unclear how this research differs from existing work. There should be an explicit section addressing the discussion of results and implications.

Thanks for your valuable suggestion. We have gone through the literature review by highlighting the differences between this study in ‘2 Literature review’ and existing studies and adding the theoretical and practical implications of this study in the ‘7 Conclusion’ section. All modifications are marked in red.

5.Comment 4,5&6 by previous reviewer, has not been addressed clearly. If the author(s) have used ‘Matlab’, describe the procedure thoroughly. 

Thanks for your valuable counsel. In this paper, in order to show the details of the decision making results of this study more intuitively, numerical analysis is used to analyse the key indicators under different covenants. With the help of Matlab software, we use the parameter variables to be analysed as independent variables and dependent variables, and then substitute the values of the rest of the parameters to get the required equations, enter the equations by means of codes in the software, set certain intervals, and draw them with different lines, and then finally run all the codes to draw the corresponding images. By observing the images, the trend of key indicators can be analysed.

6.Provide a general discussion along with subheadings elaborating on 1) theoretical implications, 2) practical implications, 3) limitations and scope for future research, and 4) provide a short conclusion at the end of the manuscript.

Thank you for your valuable suggestions! In response to your suggestions we have added and improved the corresponding paragraphs in the text, added subheadings to the corresponding content, and marked the revised draft in red.

Thank you again for your valuable and constructive comments and suggestions on our manuscript. we sincerely hope that the revised manuscript could be acceptable for you.

---

## [Decision Letter · Decision Letter 2]

14 Feb 2024

Research on technological innovation decision-making considering government subsidies and corporate reputation

PONE-D-23-30152R2

Dear Dr. Huang,

We’re pleased to inform you that your manuscript has been judged scientifically suitable for publication and will be formally accepted for publication once it meets all outstanding technical requirements.

Kind regards,

Jitendra Yadav, Ph.D.

Academic Editor

PLOS ONE

Reviewers' comments:

Reviewer's Responses to Questions

**Comments to the Author**

1. If the authors have adequately addressed your comments raised in a previous round of review and you feel that this manuscript is now acceptable for publication, you may indicate that here to bypass the “Comments to the Author” section, enter your conflict of interest statement in the “Confidential to Editor” section, and submit your "Accept" recommendation.

Reviewer #1: All comments have been addressed

Reviewer #3: All comments have been addressed

2. Is the manuscript technically sound, and do the data support the conclusions?

Reviewer #1: Yes

Reviewer #3: Yes

3. Has the statistical analysis been performed appropriately and rigorously? 

Reviewer #1: Yes

Reviewer #3: Yes

4. Have the authors made all data underlying the findings in their manuscript fully available?

Reviewer #1: Yes

Reviewer #3: Yes

5. Is the manuscript presented in an intelligible fashion and written in standard English?

Reviewer #1: Yes

Reviewer #3: Yes

6. Review Comments to the Author

Reviewer #1: Improving the research and increasing its prospects of adoption will require addressing these criticisms

Reviewer #3: (No Response)

7. PLOS authors have the option to publish the peer review history of their article (what does this mean?). If published, this will include your full peer review and any attached files.

Reviewer #1: No

Reviewer #3: No

---

## [Editor Report · Acceptance letter]

11 Mar 2024

PONE-D-23-30152R2 

PLOS ONE

Dear Dr. Huang, 

I'm pleased to inform you that your manuscript has been deemed suitable for publication in PLOS ONE. Congratulations! Your manuscript is now being handed over to our production team.

Kind regards, 

on behalf of

Dr. Jitendra Yadav 

Academic Editor

PLOS ONE